# A New Proximal Adjustable Sling ATOMS SSP^®^ Implantation Technique with Focus on the Urethral Bulb: Lessons Learned from Revision Surgery

**DOI:** 10.3390/jcm12134409

**Published:** 2023-06-30

**Authors:** Fabian Queissert, Benedict Bruecher, Andres J. Schrader

**Affiliations:** Clinic for Urology and Pediatric Urology, University Hospital Muenster, 48149 Münster, Germany

**Keywords:** male stress incontinence, ATOMS, adjustable slings, urethral bulb, urethral sphincter

## Abstract

Background: Adjustable sling ATOMS-SSP results in ventral compression of the urethra with favorable results in the treatment of men with mild to moderate stress incontinence. However, with transobturator tunneling and mesh fixation, the surgeon has a range of options, which leads to different results and sometimes unfavorable positioning of the silicone cushion. Using retrograde urethrography (RUG), we identified ATOMS patients with considerable misplacement. We then modified the implantation technique when we performed the revision, and now present here our first experiences with this new surgical technique. Methods: Patients after ATOMS-SSP implantation at our clinic were systematically subjected to a RUG if incontinence persisted after adjustments. In case of unfavorable positioning, a revision was performed with the aim of achieving an idealized urethroproximal position of the silicone pad. During follow-up, a repeat RUG was performed, and both subjective and objective outcome parameters were recorded. Results: Four men met the above criteria and underwent revision with reimplantation using our new technique. All patients postoperatively experienced significantly improved continence. RUGs demonstrated an ideal ATOMS position immediately below the proximal bulbar urethra. Conclusions: Our proximal implantation technique, presented here for the first time, allows optimal positioning of the ATOMS SSP, which is reflected in the objective parameters and RUG. Its use in primary implantation should also be considered and an expansion to the indication of severe stress incontinence seems possible, but this should only be done in studies.

## 1. Introduction

Stress urinary incontinence (SUI) is also a common complication of modern laparoscopic and robot-assisted radical prostatectomy. Depending on the definition, such a complication persists in 2–20% of patients for more than 1 year [1]. Additionally, after transurethral desobstruction, SUI occurs in 0–0.5% of cases. If the patient is sufficiently bothered by the symptoms, surgical therapy can be performed. Available options for mild to moderate stress incontinence include a fixed sling (e.g., Advance XP^®^, Boston Scientific, Marlborough, MA, USA). For severe incontinence, an artificial urinary sphincter (AUS—e.g., AMS 800^®^, Boston Scientific) is recommended. Both procedures were most recently compared in a prospective randomized study with satisfactory results ultimately observed [2]. The participating patients were more satisfied with the outcome after implantation of an AUS. However, when considering a longer follow-up period, the AUS is characterized by a relevant complication rate, so that it should be used advisedly [3]. A further therapeutic option exists in the form of adjustable slings. In addition to the Argus T^®^ (Promedon, Córdoba, Argentina) and Reemex^®^ (Neomedic, Madrid; Spain) models, the ATOMS SSP^®^ (Adjustable TransObturator Male System with Silicon covered Scrotal Port, Agency for Medical Innovations (AMI), Feldkirch, Austria) in featuring simple adjustability via a small silicone-covered titanium port is particularly popular. A meta-analysis demonstrated satisfactory continence rates (67% social continence) and improvement rates (>50% in 90% of cases) for all ATOMS generations (inguinal port IP, scrotal port SP, silicon covered scrotal port SSP) [4]. In a new prospective multicenter Canadian study, a continence rate (≤1 pad/24 h) of 73.3% 19 months after surgery was observed for the current ATOMS version SSP [5]. In addition to the published satisfied patients, however, this left more than ^1^/_4_ of patients who continued to suffer from urinary leakage after ATOMS implantation and may require revision. Angulo et al. could demonstrate for this cohort that a repeat ATOMS implantation with replacement of the system is a viable option and achieved results comparable with the secondary implantation of an AUS [6]. A too-loose or asymmetrical fit of the first ATOMS has been discussed as a possible cause for the effectiveness of a subsequent ATOMS implantation. However, recommendations on how exactly to perform the revision were not given. In our department, we have been performing the implantation of the ATOMS-SSP since 2016. As a standard procedure, we routinely perform retrograde urethrography (RUG) for filling volumes >15 mL while simultaneously partially filling the ATOMS SSP with contrast to evaluate the position of the silicone cushion and its relationship to the sphincter. Over time, it became apparent that the more proximal, i.e., closer to the sphincter, the ATOMS silicone pad was positioned, the better the continence results were after further adjustment and vice versa. This led us to conclude that revision in these patients must result in a more proximal position of the cushion to achieve the desired outcome. We modified the implantation technique accordingly at a few crucial points and want to share with you here our new surgical technique, the outcomes achieved, and the insights gained from such revised patients. This feasibility study should primarily become the starting point for a discussion on how the ATOMS should mainly be used in the future. Building on this, mainly theoretical, work, further work could follow to verify the concept and possibly establish a new standard in ATOMS implantation. 

## 2. Materials and Methods

Between 2016 and March 2021, 85 ATOMS SSP implantations were performed at Muenster University Hospital. The technique published by Seweryn et al. was used in each case [7]: A perineal midline incision of 3–4 cm was made, and the subcutaneous tissue over the penile and bulbar urethra was dissected. After reaching the bulbospongiosus muscle, the neurovascular structures containing the scrotal posterior nerves were dissected and laterally mobilized on both sides of the bulb to avoid compression of these structures by the ATOMS device and thus prevent prolonged neural irritation. Along the urethra, the inferior pubic ramus was identified, and the adipose tissue was lateralized as far as possible. The ends of the corpora cavernosa columns were exposed here, and the foramen obturatorium could be palpated laterally. After safe separation of the neurovascular structures on both sides, transobturator puncture was performed with atunneller, allowing either direct placement of the mesh or use of a placeholder ligature. The tunneller was deeply introduced parabulbar, and the mesh was thus placed transforaminally directly or indirectly via the placeholder ligature. With strong tension on the mesh arms, the cushion was placed on the urethra with the upper edge coming to rest at about the level of the puncture of the tunneller into the foramen obturator. The pulled-through mesh arms were now fixed to the silicone cushion from the dorsal side using the preattached unabsorbable sutures as manufactured. This surgical technique allowed for stable fixation of the ATOMS device on the urethra without any tendency to loosen. During surgery, the cushion was filled via the attached port with isotonic saline, de-aerated, and passively filled to achieve a steady state (usually 6–9 mL remain spontaneously in the system). After digital preparation, the port was placed in the scrotum. Closure was performed continuously using absorbable sutures. The catheter normally remained in place for 2 days, and the intraoperative antibiotic treatment with a third-generation cephalosporin continued for a total of 24 h. Patients were discharged on the 2nd–3rd postoperative day.

At 4–6 weeks follow-up, our patients were presented as outpatients. A 24-h-pad test, ICIQ-SF, PGI-I, and uroflowmetry/bladder sonography were routinely performed. If necessary and after ruling out symptomatic bladder voiding dysfunction, an adjustment was performed, with 2–5 mL of saline solution administered per outpatient visit. If there was no significant improvement in incontinence with >15 mL ATOMS volume, a RUG was performed after prior application of 2 mL of iodine contrast medium via the port to assess the device position and relationship to the urethra. If this positional check revealed an unfavorable position (e.g., incorrectly oriented towards the penile perpendicular axis of the ATOMS cushion, no support of the bulbar urethra, Figure 1), revision with repeat ATOMS implantation was indicated; if the position of the cushion was acceptable but incontinence persisted after further adjustment, implantation of an artificial sphincter was recommended.

During revision, the surgical technique of Seweryn et al. was modified to allow a more favorable location and alignment of the cushion:The perineal central tendon was transected transversely on the bulbospongiosus muscle to allow a partial retrobulbar position of the silicone pad (Figure 2);The foramen obturatorium was punctured at its mediodorsal angle, approximately 1.5 cm below the previous insertion (Figure 3). The blue helix needles were used, which have a smaller diameter than the red helix needles (Figure 4);Delivery of the tip of the tunneller paraurethrally below the hiatus urogenitalis, underneath the membranous urethra. As previously described by Rheder et al. for the fixed male Sling Advance, the entrance of the introducer needle tip into the perineal wound should be in the uppermost corner between the inferior pubic ramus and urethral bulb (Figure 5);The tunneled mesh arms were fixed using high tension to achieve an increased contact pressure of the cushion (in Cases 3 and 4) (Figure 6).

All other surgical steps and the postoperative setting were performed as described above. Postoperatively, the satisfaction (ICIQ-SF, PGI-I), objectively quantifiable continence situation (24-h pad test), and micturition conditions (IPSS [Question 3—intermittency, Question 5—weak stream, Question 6—straining] and uroflowmetry were recorded. A RUG was performed in each case to verify the position of the cushion.

## 3. Results

During the follow-up after the first ATOMS implantation, four patients were disturbed by persistent severe urinary incontinence ranging from 25 to 350 mL in the 24-h pad test at an ATOMS filling volume of 18.5–22 mL (Table 1). In all cases, an unfavorable position of the cushion was observed during the RUG with a contrasting medium-filled device: support was found more toward the proximal penile instead of the bulbar urethra as well as a siphon-like elongation of the bulbus urethrae was present (Figure 7a–d). Direct support of the external urethral sphincter muscle could not be detected. All four cases underwent revision surgery using our modified surgical technique.

During the outpatient follow-up after 4–6 weeks and any necessary further adjustments, all patients showed an improvement in continence compared to the situation with the implanted and adjusted 1st ATOMS cushion. Case 2 experienced a slight improvement in urinary incontinence. Imaging showed a rather dorsal placement of the silicone pad (Figure 7b’). We then further adapted our technique in Cases 3 and 4 and added the tight mesh fixation (Figure 6). Figure 8 shows the intraoperative image of Case 1, where the pseudocapsule of the old cushion is located distally and thus covers the penile urethra distally, while the 2nd ATOMS is approx. 1.5 cm lower/proximal immediately underneath the bulbar urethra.

For demonstration purposes, a repeat RUG was performed in our cases (Figure 7: Cases 1 and 2 showed an improved position, and in both patients, an improvement of continence could be observed). In Cases 3 and 4, where tight mesh fixation was used, good continence was achieved. In RUG of Case 3, an idealized position with the support of the proximal bulbar urethra was observed (Figure 7c’ and Figure 9). In Case 4, the patient was fully satisfied so no adjustment and consecutively no postoperative RUG was performed.

## 4. Discussion

Since 2009, the ATOM System has been used for the treatment of mild to moderate stress incontinence, but assumptions about its functioning have only been made in recent years. Virseda et al. and Queissert et al. have shown that the unilateral compression of the urethra does not result in obstructive micturition at normal adjusted volumes, and they concluded that the urethra can deflect in other directions due to the unilateral compression [8,9]. However, they did not provide any evidence for the assumption that indirect support of the membranous sphincter results from this. Nevertheless, Queissert et al. support Rheder et al.’s opinion that the urethral bulb has significant importance for continence [10]. Rheder stated that the bulbospongiosus muscle also plays a role in continence: Contraction of the bulbospongiosus muscle will increase the pressure within the corpus spongiosum, which in turn will transfer the pressure wave onto the urethral wall. Based on this hypothesis, Rheder et al. argued that in addition to the cranial displacement of the membranous urethra in patients after implantation of a fixed male sling, a second effect could exist: the retrobulbar sling-loop would serve as a hammock. During physical activity, increased blood flow in the corpus spongiosum causes swelling of the distal rhabdosphincter, thus extending its coaptation zone. Rheder assumed that a coaptation zone of the rhabdosphincter of >1–1.5 cm is needed for achieving continence. For the adjustable sling ATOMS, this has two consequences: 1. it should work better the more elastic and intact the corpus spongiosum, and 2. it should work better the closer the cushion is to the urethra and to the rhabdosphincter. Point 1 was already demonstrated by Ruiz et al. [11]. They investigated a small ATOMS cohort using intraurethral pressure measurement during ATOMS surgery: patients with a higher elasticity of the urethral bulb had a higher likelihood of achieving social continence compared to males with a more rigid bulb. In this study, we now describe for the first time Point 2, a correlation between the positioning of the silicone cushion and continence improvement. A more distally located cushion results in a worse continence outcome. Therefore, the fact that the surgeon had multiple approaches available in the ATOMS implantation must be noted as problematic. In particular, transobturator tunneling with mesh placement and cushion positioning results in a high degree of positional variability in multidimensional space due to the width of the foramen obturatorium and varied exit points of the transobturatorily placed helix needle in the perineum and variations in mesh attachment. Especially novice users often struggle with adequate positioning. The classic transobturator implantation technique of the female TVT-O sling or Advance XP fixed sling involves puncturing the obturator foramen at the medioventral margin, approximately two finger widths below the attachment of the tendon of the adductor longus muscle. The puncture height is irrelevant for the TOT procedures, as the tension of the sling (TVT-O, Advance XP) or the paraurethral exit of the helix needle (Advance XP) determines the function/success of the surgery. In contrast, in ATOMS the entry point of the tunneller determines the final position of the silicone pad. Pulling on the caudally led-out mesh arms places the upper edge of the device on a horizontal plane with the upper transobturator insertions. With the conventional TOT puncture technique, there is, thus, a risk, as in our four cases, of the silicon pad being positioned too far distally under the penile urethra. Having identified this problem with the help of RUG, we modified our surgical technique. Transecting the central perineal tendon on the bulbospongiosus muscle is easier compared to transecting between the urethra bulb and bulbospongiosus muscle, as is necessary when implanting the Advance XP [12]. In our cases, the connection consisted of only a narrow tendon strand that ran sagittal and was easily incised horizontally. This approach allows mobilization of the urethral bulb with three effects: first, it allows dorsoproximal positioning of the cushion; second, as with the fixed sling, it can result in a shift of the membranous urethra towards the intrapelvic region; and third, it allows safe separation and lateralization of the posterior scrotal nerves. As a further modification, the obturator foramen is punctured at a mediodorsal angle, approximately 1.5 cm below the usual TOT tunneling. Using the blue tunnelers (Figure 4) with a small diameter, safe lead-out is achieved directly under the urogenital hiatus lateral to the urethra, a localization already used in the outside-in technique for placing the Advance XP [12]. The upper edge of the silicone cushion thus comes to rest more proximally, namely below the transition from penile to the bulbar urethra, after tension is applied to the extended mesh. Depending on the individual anatomy of each patient, excessively posterior placement may occur during dorsal puncturing of the obturator foramen, as demonstrated in Case 3 in the RUG (Figure 7c’). This may result in inadequate support of the urethral bulb and persistent incontinence. To avoid this problem, we added another surgical step, which ultimately resulted in the perfect placement of the pad. The mesh arms are fixed with increased tension and under intense pressure on the lower edge of the silicone cushion in order to shift the device in the direction of the proximal bulbar urethra. This prevents a too-posterior position (as in Case 2) and aligns the perpendicular pressure point ideally towards the membranous urethra. In addition to providing direct support to this continence-relevant region, there is also less need for adjustments, as the silicone cushion is positioned closer to the urethra. Postoperative RUG of Case 3 in particular demonstrated the effectiveness of our method (Figure 9). In Case 4, also operated with the augmented tight fixation of the mesh on the cushion, adjustment with contrast medium and subsequent retrograde urethrogram (RUG) was not done at the patient’s request, despite social continence. Our surgical technique allows for compressing the corpus spongiosum more effectively compressed, thus adopting and extending the effect of the fixed sling described by Rehder et al. by the option of adjustability. In addition, there is now a relatively small distance to the urethral sphincter when aligned perpendicularly, which could result in direct support of the rhabdosphincter. None of our patients complained of any restriction of micturition; rather, our technique seems to preserve the natural micturition process, as the urethra no longer seems to be constricted between the symphysis and the silicone cushion. It seems our surgical technique does not increase the ATOMS risk profile, rather, it could reduce the probability of having to do revision surgery for persistent incontinence. Should a revision be necessary after all, another advantage can be recognized: the proximal position of the cushion under the bulbar urethra leaves the proximal penile and parts of the distal bulbar urethra unencumbered, which should simplify AUS implantation if needed.

This work is particularly limited by the small number of patients at a single center. Therefore, the work has hereby rightly earned the designation of a feasibility study. However, with a high level of experience of >250 implanted ATOMS and >150 AUS, the authors are certainly able to evaluate clinical courses. With the knowledge gained with this technique and the feedback from patients, a revival of the conventional surgical technique no longer seems conceivable in our clinic.

## 5. Conclusions

Using RUG, we succeeded for the first time in describing the relationship between ATOMS positioning and patient outcome. With the aim of optimizing the position of the silicone pad, we modified the ATOMS implantation technique: a transverse incision of the perineal central tendon on the bulbospongiosus muscle, a deeper puncture of the obturator foramen, a lead-out of the blue helix needles in the urogenital hiatus, and a tight fixation of the mesh resulted in a significant improvement of the continence situation in the patients revised in this way. A new RUG demonstrated a more proximal position of the silicone cushion with ideal support of the urethra bulb, which may lead to a longer zone of coaptation of the rhabdosphincter. There does not appear to be any increased risk profile or impairment of micturition. On the contrary, both the assumably less frequent need for adjustment and the preservation of the proximal penile urethra and, consecutively, the option of implanting an artificial sphincter in the future are further advantages. The use of this surgical technique is now also being evaluated in the primary situation within the framework of a prospective observational study and is likely to be adopted in this indication as well. The remarkable results of ATOMS so far could be further improved; an extension to patients with severe stress incontinence is conceivable but should initially also only take place in a study context.

## Figures and Tables

**Figure 1 jcm-12-04409-f001:**
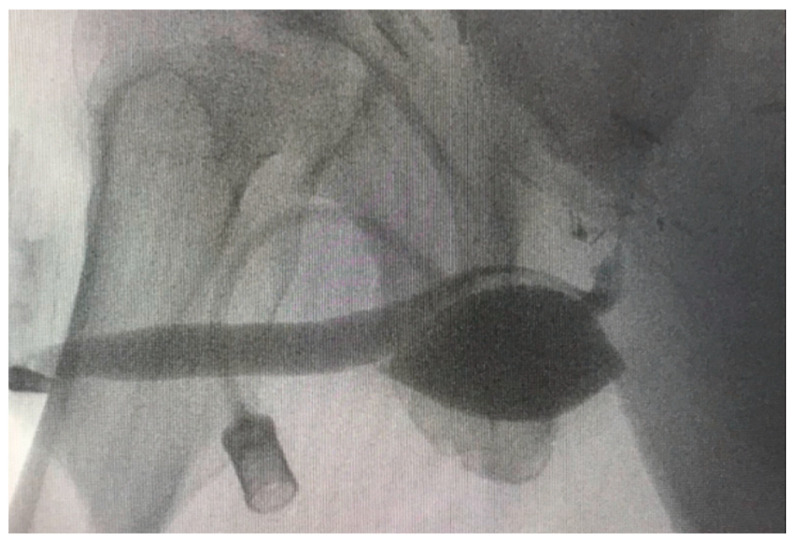
RUG of Case 4 with further incontinence after the first ATOMS SSP (filled with 16 mL) revealed a too-distal positioning of the cushion. This results in a compression of the penile urethra, with the sphincter being located at a considerable distance and presumably unable to be directly or indirectly supported.

**Figure 2 jcm-12-04409-f002:**
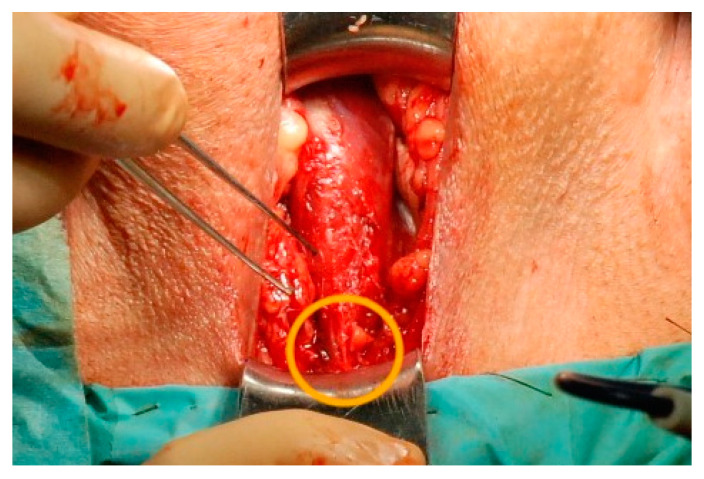
Perineal wound, the bulbospongiosus muscle is exposed, caudally the central perineal tendon can be identified (orange circle), which can be easily transversely incised, thus mobilizing the bulb and creating space for a silicone pad.

**Figure 3 jcm-12-04409-f003:**
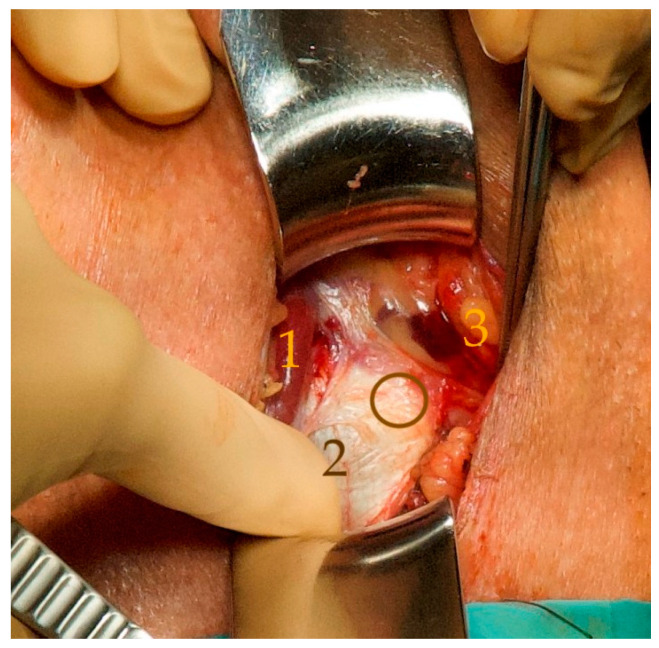
The bulb is medialized with the bulbospongiosus muscle (1). This allows the proximal extension of the left corpus cavernosum to be identified on the inferior ramus of the pubic bone (2). The finger passes beneath the palpable lower medial edge of the pubic bone. At a 45° angle lateroventral to this edge, a gap in the obturator foramen is palpable, which can be punctured with the tunneler (circle). The vascular-nerve bundle is laterally displaced to prevent compression of the nervi scroti posterior and thereby avoid prolonged pain (3) (for better visibility, we used a picture of a patient with the first ATOMS SSP implantation).

**Figure 4 jcm-12-04409-f004:**
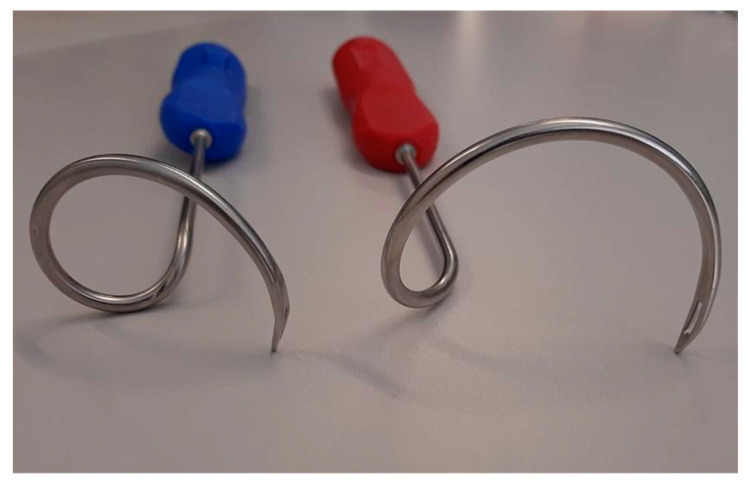
A.M.I. Gmbh distributes different tunnellers. For the proximal implantation, the blue one enables the passage of the foramen obturatorium in a confined space. This helps to protect surrounding structures such as the rectum and enables ideal placement of the needle tip and consecutively of the mesh arms in the urogenital hiatus (Figure 5). The use of the red tunneller is also possible but requires special protection of the rectum during the passage of the foramen by manual dorsal compression.

**Figure 5 jcm-12-04409-f005:**
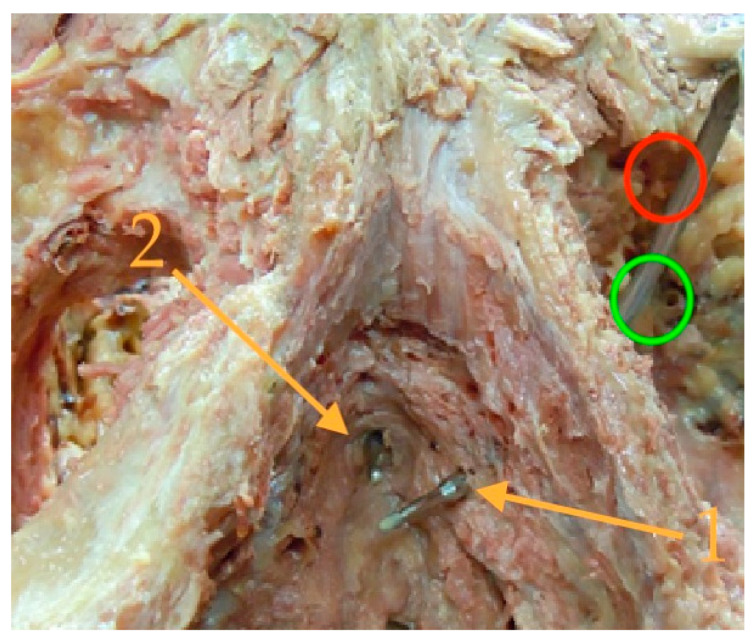
(Picture with kind permission of Dr. Rheder, Innsbruck) The prepared pelvis demonstrates the ideal position of the tip of the tunneller: The insertion point (green circle) is located ventrolateral of a small palpable pubic edge (the red circle marks the usual TOT-needle insertion which is not useful for ATOMS placement). The needle tip should be exited in the urogenital hiatus (1), lateral and below the membranous urethra (2). For this purpose, the grip hand should be slightly lowered after perforation of the outer and inner membrane of the obturator foramen. The same needle positioning is also recommended for the fixed male sling.

**Figure 6 jcm-12-04409-f006:**
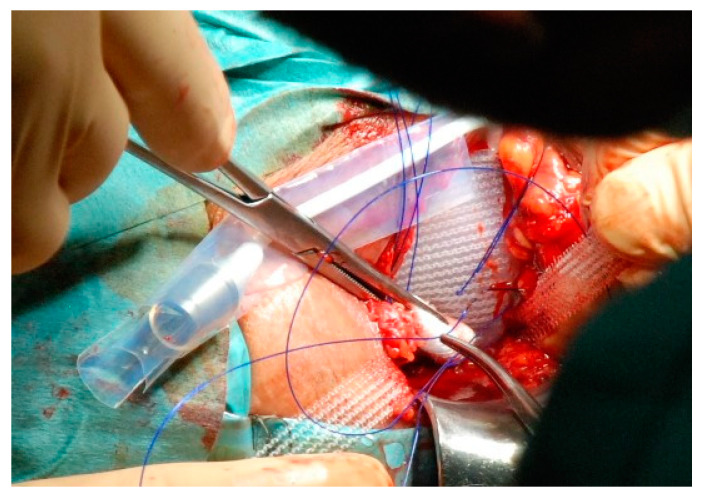
When fixing the mesh, care should be taken to ensure that the tension is as high as possible. For this purpose, we use a cotton swab to press the lower edge of the silicone pad behind the urethra bulb. At the same time, the prefixed sutures are pulled through the pulled mesh at a deep point. With correct application, the lower edge of the silicone cushion is slightly shifted towards the pelvis. This results in partial retrobulbar support.

**Figure 7 jcm-12-04409-f007:**
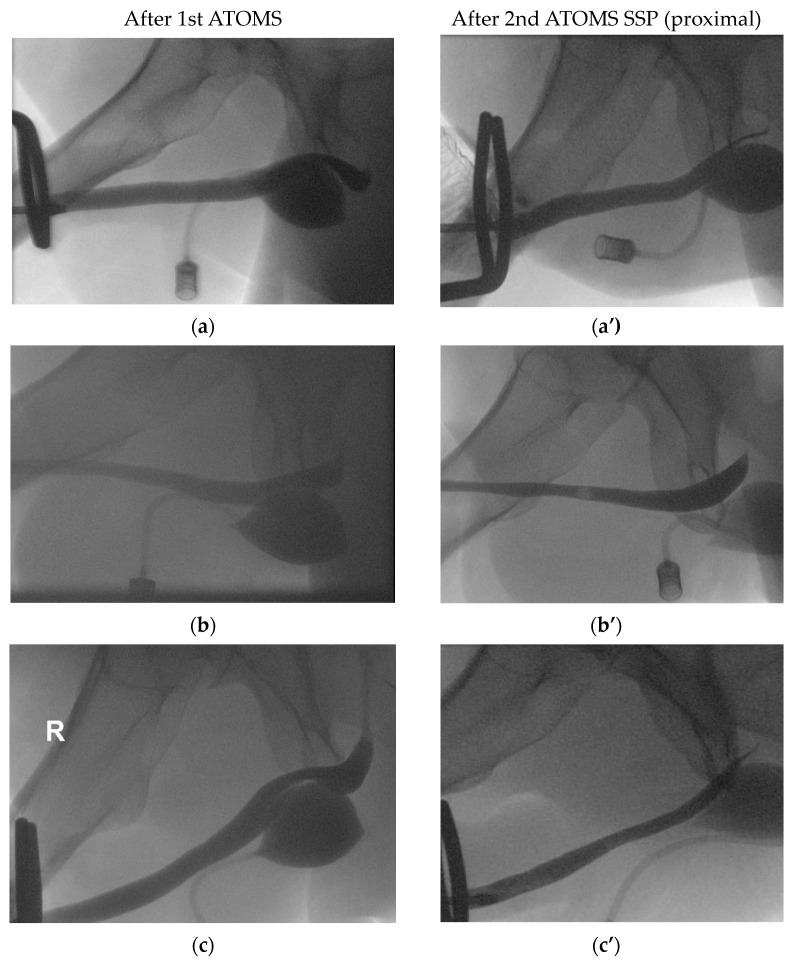
(**a**) Case 1, 1st ATOMS SSP with compression of the penile urethra and a siphon-like bulbar elongation. (**a’**) Case 1, 2nd ATOMS SSP proximal with direct compression of the bulbar urethra results. (**b**) Case 2, 1st ATOMS SSP with a too distal placement of the silicone pad under the proximal penile urethra. (**b’**) Case 2, 2nd ATOMS SSP proximal with a too dorsal placement which explains the only partial improvement. We then further adapted our surgical technique and included the tight mesh fixation in Cases 3 and 4. (**c**) Case 3, 1st ATOMS SSP with too distal compression of the penile urethra and a siphon-like elongation of the bulbar urethra. (**c’**) Case 3, 2nd ATOMS SSP proximal with an ideal placement (see also Figure 9). (**d**) Case 4, 1st ATOMS SSP with too distal compression of the penile urethra. No postoperative RUG was available due to the high satisfaction of Case 4.

**Figure 8 jcm-12-04409-f008:**
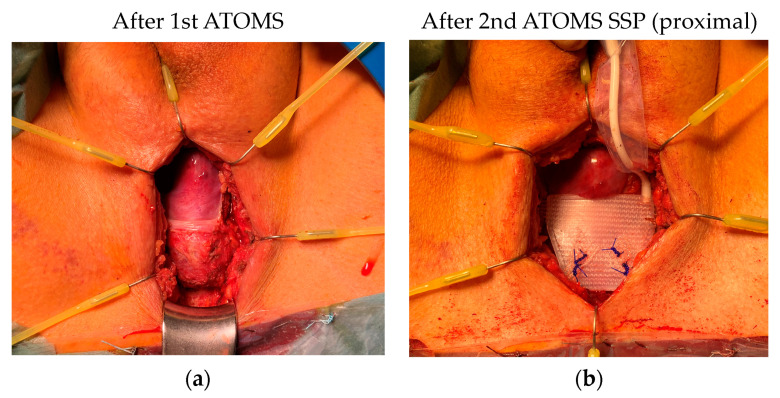
(**a**) Case 1, perineal wound with removed 1st ATOMS SSP. The pseudocapsule covers the more distal bulbar/proximal penile urethra. (**b**) Case 1, 2nd ATOMS SSP is fixed and covers directly the urethral bulb (mesh arms were not yet fixed tight).

**Figure 9 jcm-12-04409-f009:**
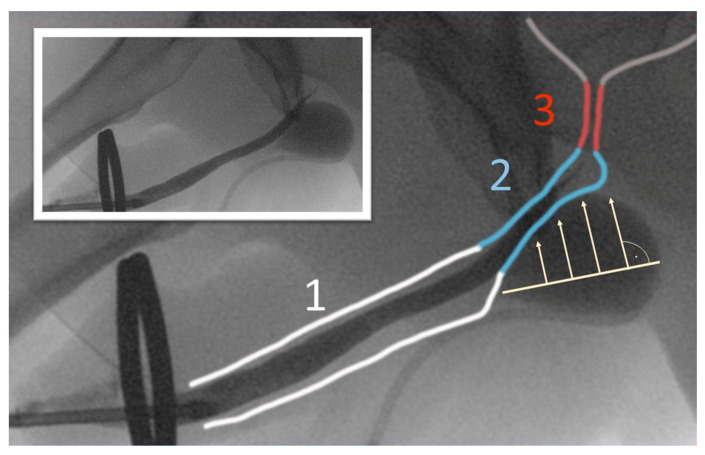
(1 penile urethra, 2 urethral bulb, 3 rhabdosphincter/sphincteric coaptation zone). RUG of Case 3 after 2nd ATOMS SSP using our proximal implantation technique. ATOMS SSP proximal now allows a direct compression of the urethral bulb and also possibly of the rhabdosphincter. A lengthening of the coaptation zone can be assumed. A complete dryness resulted in this case.

**Table 1 jcm-12-04409-t001:** Basic data and Outcome after 1st and 2nd ATOMS-SSP Surgery.

	Case 1	Case 2	Case 3 *	Case 4 *
Basic data				
age at 1st ATOMS-surgery	54	68	68	59
reason for SUI	RARP	RARP	Open RPx	Open RPx
Time of surgery	19 April	17 May	January 2012	July 2019
Irradiation	No	No	No	No
*Incontinence before 1st ATOMS*				
ICIQ-SF	4,4,5 = 13/21	5,6,8 = 19/21	5,6,7 = 18/21	5,4,10 = 19/21
Safety pads/24 h (*n*)	6	3	7	7
Urine loss/24 h (g)	700	180	210	600
*Urodynamic evaluation*				
cystometric volume (mL)	n.a.	508	483	790
Detrusor overactivity	n.a.	No	No	No
Acontractile detrusor	n.a.	Yes	No	No
Residual urine (in mL)	0	0	0	150
**1st ATOMS SSP**				
Date of surgery	20 May	20 July	20 February	20 July
Initial filling volume (mL)	7.2	6.0	7.2	7.0
No. of Adjustments (*n*)	6	4	5	5
Max. ATOMS filling volume (mL)	23	19	19	21
*last uroflowmetry*				
voided volume (in mL)	84	278	495	528
Q_max_ (in mL)	9.4	32	17	23.1
residual urine (in mL)	10	34	0	200
*Incontinence before 2nd ATOMS*				
ICIQ-SF	4,4,5 = 13/21	4,4,5 = 13/21	4,6,8 = 18/21	4,4,7 = 15/21
PGI-I (compared to pre surgery)	4 (no change)	4 (no change)	4 (no change)	3 (little better)
Safety pads/24 h (*n*)	4	2	2	2
Urine loss/24 h (g)	350	40	100	75
**2nd ATOMS SSP (proximal)**				
Date of surgery	21 March	22 January	22 August	23 March
Initial filling volume (mL)	7.5	6.5	7.0	6.5
Follow up (months)	25	17	9	2
No. of Adjustments (*n*)	3	6	3	0
Max. ATOMS filling volume (mL)	19.5	25	14.5	6.5
*last uroflowmetry*				
voided volume (in mL)	566	323	333	331
Q_max_ (in mL)	18.6	32.6	49.4	19.5
residual urine (in mL)	10	59	0	180
*Incontinence before 1st ATOMS*				
ICIQ-SF	4,4,6 = 14/21	4,4,5 = 13/21	0,0,0 = 0/21	1,2,2 = 5/21
PGI-I (compared to 1st ATOMS)	2 (much better)	3 (little better)	1 (very much better)	1 (very much better)
Safety pads/24 h (*n*)	4	2	0	1
Urine loss/24 h (g)	46	20	0	13

* During implantation of 2nd ATOMS all surgical steps inclusive of tight connection of mesh/silicon pad were used (Figure 6). RARP = robotic-assisted radical prostatectomy, RPx = radical prostatectomy, ICIQ-SF = International Consultation on Incontinence Questionnaire-Short Form, PGI-I = Patient Global Impression of Improvement.

## Data Availability

Additional data not presented in this study are available on request from the corresponding author. The data are not publicly available due to privacy.

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
