# Peer review of "A New Proximal Adjustable Sling ATOMS SSP® Implantation Technique with Focus on the Urethral Bulb: Lessons Learned from Revision Surgery"

_jcm, 2023, doi:10.3390/jcm12134409_

Round 1
Reviewer 1 Report
Stress urinary incontinence is a common complication following prostatectomy that requires exceptional management.
There are some minor corrections that need to be made before publication of the manuscript.
As a first step, the purpose of the study should be stated in the last paragraph of the Introduction.
Furthermore, the authors did not mention any limitations to the study, such as the limited number of patients or the limited experience of the single center.
The manuscript need modified and corrected in terms of grammar and English.
Minor editing of English language required
Author Response
Dear Reviewer, Thank you for your efforts in reviewing our manuscript. In particular, we thank you for your wise and constructive comments.
We hope that the additions in the introductory section and the descriptions of the absolutely existing limitations meet with your approval.
Regarding the English language review, we will consult a native English speaker before the final revision of the manuscript. Unfortunately, due to the short time to respond to the review, we have not been able to do so so far. The revised version will now be sent for English correction in parallel with the upload to MDPI. If desired, the final version can be shared again.
Once again, we thank you very much.
Best regards
Reviewer 2 Report
The paper presented to us has merit and is an excellent starting point for a scientific discussion on the surgical treatment of SUI in men using the ATOMS system. The authors present a hypothesis based on previous studies by other groups on urethral and sphincter support.
The surgical approach presented in the ATMOS surgical revision process is a reinterpretation of the original technique described for its placement. The anatomical theory and the images presented are innovative in this field and could be an added value in the treatment of this type of patients. It could be the beginning of a line of research to optimize the use of ATOMS in cases of refractory incontinence. However, as the authors announce, it lacks validation in the context of clinical trials.
However, the series presented is small (only 4 cases), of which 2 are well documented and with good results. However, case 2 did not show signs of significant improvement and case 4, although it improved, is not properly documented from the imaging point of view, for reasons beyond the control of the authors.
So, here are two points. First, I would like case 2 to be discussed in the text. Looking to the image, it seems that the placement was excessively posterior, not achieving the intended effect of sphincter support. The second point, for editorial decision, since only 4 cases are presented, with only 2 successful and well documented, the article base most of its worth on the theoretical proposal, then on its intrinsic results.
Author Response
Dear Reviewer,
Thank you for your efforts in reviewing our manuscript. In particular, we thank you for your wise and constructive comments. You are absolutely right about case 2, we noticed the overcorrection and a mere halving of the urine loss did not lead to a significant improvement of the quality of life. We have expanded and improved the relevant section in the discussion and hope to have adequately corrected your point of criticism.
Once again, we thank you very much.
Best regards